# Mechanisms of and Potential Medications for Oxidative Stress in Ovarian Granulosa Cells: A Review

**DOI:** 10.3390/ijms24119205

**Published:** 2023-05-24

**Authors:** Siheng Liu, Yunbing Jia, Shirui Meng, Yiran Luo, Qi Yang, Zezheng Pan

**Affiliations:** 1Queen Mary School, Jiangxi Medical College, Nanchang University, Nanchang 330006, China; drrocker@163.com (S.L.);; 2College of Basic Medical Sciences, Jiangxi Medical College, Nanchang University, Nanchang 330006, China

**Keywords:** oxidative stress, reactive oxygen species, granulosa cell, premature ovarian failure, polycystic ovary syndrome, apoptosis

## Abstract

Granulosa cells are essential for follicle initiation and development, and their abnormal function or apoptosis is a crucial factor leading to follicular atresia. A state of oxidative stress occurs when the balance between the production of reactive oxygen species and the regulation of the antioxidant system is disturbed. Oxidative stress is one of the most important causes of the abnormal function and apoptosis of granulosa cells. Oxidative stress in granulosa cells causes female reproductive system diseases, such as polycystic ovary syndrome and premature ovarian failure. In recent years, studies have confirmed that the mechanism of oxidative stress in granulosa cells is closely linked to the PI3K-AKT signaling pathway, MAPK signaling pathway, FOXO axis, Nrf2 pathway, NF-κB signaling pathway, and mitophagy. It has been found that drugs such as sulforaphane, *Periplaneta americana* peptide, and resveratrol can mitigate the functional damage caused by oxidative stress on granulosa cells. This paper reviews some of the mechanisms involved in oxidative stress in granulosa cells and describes the mechanisms underlying the pharmacological treatment of oxidative stress in granulosa cells.

## 1. Introduction

Granulosa cells (GCs) play an essential role in follicular development. They are responsible for converting androgens to estrogens, synthesizing progesterone, and secreting insulin-like growth factor (IGF) [1,2]. As the supporting cells of female gametes, GCs provide most of the nutrients required for their growth and development [3]. In addition, GCs and oocytes can exchange substances and signals through gap junctions. Their interactions regulate follicle maturation, ovulation, and fertilization [4]. During follicular development, the proliferation of GCs and the secretion of follicular fluid contribute to the formation of sinusoidal follicles or Graafian follicles. Early sinusoidal follicles produce anti-mullerian hormone (AMH), inhibiting the activation of primordial follicles [5,6]. During the luteal–follicular transition phase of each menstrual cycle, follicles undergo cyclic recruitment and then perform the dominant follicle selection mechanism, with most sinusoidal follicles undergoing atresia [7]. GCs have been shown to play an essential role in determining the fate of the follicle, and GC apoptosis is a critical factor in follicular atresia [8]. It was found that no apoptotic cells were observed in growing healthy follicles; apoptotic cells appeared in early atretic follicles and gradually increased in number, while in progressive atretic follicles, a majority of GCs underwent apoptosis [9]. Additionally, the potential cell damage caused by oxidative stress (OS) is one of the reasons for GC apoptosis.

OS is a disturbance in prooxidant–antioxidant balance and tends toward an oxidative state. Decreased levels of antioxidants and increased reactive oxygen species (ROS) can cause OS. The effects of OS depend on the magnitude of these changes, with severe OS leading to cell death [10]. The antioxidant systems present in the body can be divided into two categories: enzymatic antioxidant systems, including superoxide dismutase (SOD), catalase (CAT), glutathione peroxidase (GSH-Px), etc.; and non-enzymatic antioxidant systems, including ergothioneine, vitamin C, vitamin E, glutathione, melatonin, carotenoids, etc. Antioxidants play an important role in the regulation of OS. They can scavenge excess ROS and help maintain the oxidant–antioxidant balance of the body, thus inhibiting the occurrence of OS [11]. In contrast, excessive production of ROS affects the structure and function of nucleic acids, lipids, and proteins, thus promoting OS. ROS include superoxide radical, hydroxyl radical, and hydrogen peroxide, etc. Many metabolic processes in organisms can spontaneously produce ROS, and these are referred to as endogenous ROS [12]. Exogenous physical factors or compounds, such as smoking, UV radiation, high partial pressure oxygen, hypoxia, drugs, heavy metals, environmental pollutants, etc., can also induce cells to produce ROS [13]. GCs are sensitive to ROS, and excessive ROS production may exceed the limits of the antioxidant defense system and lead to OS, resulting in GC dysfunction and even apoptosis [14,15]. ROS can inhibit GC proliferation and trigger cell death by activating pathways such as mitogen-activated protein kinase (MAPK) and c-Jun N-terminal kinase (JNK) [16,17]. The abnormal apoptosis and dysfunction of GCs caused by OS can lead to a series of pathological changes in the ovary, which may result in diseases such as premature ovarian failure (POF), polycystic ovary syndrome (PCOS), and endometriosis, affecting the fertility of the patient [18]. POF is characterized by low estrogen and high gonadotropins; and secondary amenorrhea, often with perimenopausal symptoms; it can also lead to infertility [19]. The prevalence of POF in women aged 20, 30, and 40 years is 0.01%, 0.1%, and 1%, respectively [20]. It is believed that genetic factors, environmental toxins, and immune factors cause POF [21]. Although the exact mechanisms responsible for POF are still unknown, numerous studies have shown that POF occurs with the apoptosis of GCs, suggesting a direct relationship between apoptosis of GCs and POF. Factors such as OS may lead to abnormal GC function and even apoptosis, accelerating follicular atresia, which is an important factor in triggering POF [22,23]. Patients with POF usually have lower than normal serum AMH, and there may be a significant correlation between AMH levels and 15 or more follicles in their ovaries [24]. PCOS is characterized by hyperandrogenemia, ovulatory dysfunction, and polycystic ovaries [25]. Symptoms of PCOS include menstrual sparing or amenorrhea, chronic anovulation, infertility, hirsutism, and acne [26]. The prevalence of PCOS ranges from 8 to 13% worldwide [27]. The etiology of PCOS is complex, and the pathogenesis is still not fully understood, but it is related to GCs [28]. The rate of GC apoptosis is significantly higher in rats with PCOS, and GC apoptosis is one of the causes of hyperandrogenemia. In the hyperandrogenic microenvironment, androgens act through androgen receptors to inhibit GC proliferation and promote apoptosis [29,30]. It has also been shown that AMH is involved in the pathogenesis of PCOS and that GCs are closely associated with its secretion and action [31]. The signaling pathways involved in GC OS leading to POF and PCOS may include the PI3K-AKT signaling pathway, FOXO1 pathway, p53/SIRT1 pathway, NF-κB signaling pathway, AMPK/Nrf2 pathway, Nrf2/ARE pathway, HIF-1α/BNIP3, etc. [32,33,34,35,36]. Various modifiable factors cause OS. Its progression can be inhibited with antioxidants to reduce the level of OS in GCs and thus reduce the incidence of disease. It has been found that many drugs mitigate the negative effects of OS on GCs by reducing ROS levels, increasing the expression of antioxidant genes, and inhibiting abnormal mitochondrial function and pro-inflammatory cytokines. This review aims to discuss the mechanisms and signaling pathways associated with OS in GCs and to summarize drugs with potential therapeutic value.

## 2. Mechanisms of Oxidative Stress in Granulosa Cells

OS causes disruption of some signaling pathways in GCs that control cell proliferation, autophagy, and apoptosis. Figure 1 shows the intracellular mechanism by which ROS causes OS in GCs.

### 2.1. PI3K/AKT

The phosphatidylinositol 3-kinase (PI3K)/protein kinase B (AKT) signaling pathway is a significant signaling pathway in cell signaling. mTOR is a common downstream protein of AKT, which can integrate various extracellular signals such as nutrients, energy, and growth factors, and participates in gene transcription, protein translation, ribosome synthesis, and other biological processes [37]. It also plays an important role in cell growth, apoptosis, autophagy, and metabolism [38]. In OS of GCs, the regulation of cell survival by AKT also involves the direct inhibition of FOXO phosphorylation [39]. Meanwhile, the PI3K/AKT pathway can regulate the expression of the nuclear factor erythroid 2-related factor 2 (*NFE2L2*), which in turn regulates GC proliferation [40].

RAS protein is a kind of small molecule GTPase, a crucial member of the RAS signaling pathway. There are three main subtypes of K-RAS, H-RAS, and N-RAS in the human body. RAS proteins regulate a variety of important biological events in cells, including cell proliferation, differentiation, apoptosis, and metabolism. Common downstream pathways of this protein include MAPK, BRAF (v-Raf murine sarcoma viral oncogene homolog B), and PI3K/AKT. Studies have shown that in human malignant tumors, the incidence of RAS protein mutations is quite high, especially in ovarian cancer; the mutation status of KRAS codons 12, 13, and 61 ranges from 6 to 65% in different histo-types [41,42].

The PI3K/AKT signaling pathway regulates cell growth and proliferation, and OS in GCs often results in the downregulation of the PI3K/AKT signaling pathway, leading to impaired growth and proliferation of GCs, which can affect ovarian function and cause various diseases.

In a recent study, a combined transcriptomic and metabolomic analysis of heat-stressed bovine GCs in vitro revealed that OS also occurs in GCs that have undergone heat stress, with the PI3K/AKT/mTOR pathway becoming significantly disturbed. Moreover, cAMP, AMPK, and steroid synthesis pathways are also involved [43]. PI3K/AKT/mTOR pathway downregulation can also occur in rat ovarian GCs with H_2_O_2_-induced autophagy and apoptosis, accompanied by elevated intracellular levels of ROS, MDA, SOD, GSH-Px, and CAT, which is counteracted by morroniside and inhibits caspase-3 (CASP-3) activity, thereby suppressing apoptosis [44]. In contrast, in OS induced by the environmental endocrine disruptor nonylphenol (NP), GCs’ viability in rats decreased, and apoptosis and autophagy significantly increased. During this process, the production of ROS, which is associated with the activation of the AKT/AMPK/mTOR pathway, was significantly increased. The increased risk of POF in young female cancer patients receiving cyclophosphamide chemotherapy may be associated with the activation of the PTEN/AKT/FoxO3a pathway [45]. In this pathway, PTEN catalyzes the conversion of PIP3 to PIP2, which terminates the PI3K signaling pathway and thus promotes mitochondrial-driven apoptosis in GCs. Maternal intake of particulate matter 2.5 also activates the PI3K/AKT/FOXO3a pathway and NF-κB signaling pathway, leading to reduced ovarian reserve capacity in offspring mice by promoting activation of primordial follicles with GC apoptosis [46]. Cadmium (Cd) is a heavy metal that induces ROS production and upregulates inflammatory factors and ASK1, JNK, p38, and TAK1 expression; downregulates RSK1 and RHEB expression; and inhibits ERK1/2, mTOR, and P70S6K1 phosphorylation, thereby blocking cell cycle progression and promoting apoptosis. Cd is also involved in the PTEN/AKT/FOXO3a pathway, which induces chicken follicular GC apoptosis, causing ovarian damage and follicular atresia [47]. Benzo(a)pyrene (B(a)P) is an organic pollutant that can disrupt steroid homeostasis. Additionally, benzo(a)pyrene-7,8-dihydrodiol-9,10-epoxide (BPDE) is a metabolite from B(a)P that affects the luteinization of granulosa cancer cells (KK-1), affecting the luteinization process. B(a)P or BPDE can also induce OS in GCs by impairing their antioxidant capacity and interfering with the PI3K/AKT/GSK3β signaling pathway [48]. Changes in N6-methyladenosine (m6A) methylation-related enzymes were associated with Cd-induced GC damage. In contrast, obesity-associated protein (FTO) may regulate the expression of MAX network transcriptional repressor (MNT) through m6A modification. Overexpression of FTO alleviated Cd-induced apoptosis and OS by activating the AKT/Nrf2 pathway and decreasing the expression of BAX and CASP-3 [49].

In pharmacological treatment, growth hormone can counteract the accumulation of ROS by activating the PI3K/AKT signaling pathway and the apoptosis of GCs of patients with PCOS by downregulating CASP-3 and caspase-9 (CASP-9). A significant decrease in the level of FOXO1 accompanies this process [50]. In contrast, vitamin E can improve this aspect through the cell cycle and apoptosis-related genes, such as upregulating CCND1 and Bcl-2 levels and downregulating p21, BAX, and CASP-3 levels, thus promoting bovine GC growth and inhibiting apoptosis [40]. GCs can also undergo excessive autophagy in the presence of OS, leading to PCOS [51]. In contrast, metformin ameliorates excessive GC autophagy and alleviates PCOS in rats by activating the PI3K/AKT/mTOR pathway [52]. To counteract OS, astaxanthin and metformin treat PCOS in a prepubertal female BALB/C mice model by increasing AKT expression in GCs [53]. Baicalin was found to enhance ovarian function in aged mice by activating the PI3K/AKT/mTOR signaling pathway, upregulating the expression of anti-apoptosis protein Bcl-2, and reducing the levels of pro-apoptotic proteins BAX and CASP-3, ultimately leading to reduced cell death and increased secretion of steroid hormones. As a result, baicalin shows potential for restoring ovarian function [54]. Isorhamnetin, on the other hand, promotes porcine GC proliferation and restores estrogen biosynthesis by activating the PI3K/AKT signaling pathway. It also upregulates the protein expression of Cyclin B, Cyclin D, Cyclin E, and Cyclin A, thus increasing the proportion of S-phase cells in response to the proliferation of porcine GCs [55].

### 2.2. MAPK

The MAPK signaling pathway is one of the crucial pathways in the eukaryotic signaling network. It is a critical signaling pathway for cell proliferation, differentiation, apoptosis, and stress response under normal and pathological conditions [56]. MAPK can be divided into four subfamilies: ERK, p38, JNK, and BMK1 (ERK5), representing the four classical MAPK pathways. Under normal conditions, mTOR inhibits phosphorylates Atg13 after being activated in the AKT and MAPK signaling pathway, preventing it from interacting with ULK1 and ULK2, and the autophagic response is suppressed [57]. In OS, ROS activates the ASK1 signalosome in the MAPK pathway, which activates its downstream JNK and p38-MAPK pathways, further activating p53 and FOXO, and thereby activating and inhibiting its downstream proapoptotic and antiapoptotic molecules and their regulators [58,59,60].

Transcriptome analysis of heat stress conditions in bovine GCs revealed that the occurrence of OS in GCs involves the downregulation of MAPK, which inhibits GC proliferation and stimulates apoptosis [61,62]. Overexpression of microRNA-146b-5p can lead to GC senescence and promote POF in mice by activating the Dab2ip/Ask1/p38-Mapk signaling pathway and phosphorylating γH2A.X [63]. The MAPK pathway is also involved in the effects of Cd ions on chicken follicular GCs. Excessive Cd decreases the mRNA levels of JNK and p38 and inhibits the phosphorylation of ERK1/2 proteins, blocking the cell cycle and cell proliferation and promoting apoptosis [47]. Additionally, advanced oxidation protein products can induce cell cycle arrest in human ovarian granulosa tumor cells (KGN cells) via the ROS-JNK/p38 MAPK-p21 pathway, organizing the cell cycle in the G1/G0 phase [17]. In pharmacotherapy, morroniside can inhibit human GC apoptosis by regulating MAPK signaling pathways, i.e., the p38 and JNK pathways, which regulate apoptosis-related genes *BAX*, *Bcl-2*, *CASP-9*, and *CASP-3* [64].

### 2.3. FOXO Axis

The forkhead box transcription factors class O (FOXO) is an essential regulator of various intracellular processes. FOXO1, FOXO3a, and other family members are involved in regulating important processes such as apoptosis, cell cycle, stress resistance, glucose and lipid metabolism, and inflammation [65]. ROS and other stress stimuli capable of producing ROS can regulate FOXO activation and gene expression at multiple levels [66]. On the one hand, FOXO expression is regulated at the transcriptional level, such as by acting on the upstream regulatory protein of FOXO-like p53 and affecting the regulation of its mRNA by miRNA and HuR. On the other hand, it can also affect the activity, stability, and binding of FOXO to DNA through phosphorylation and acetylation. In addition, ROS can also regulate the activation of FOXO proteins by affecting *FOXO* transcriptional co-activators. In some pathological conditions related to OS, FOXO overactivation and overexpression can lead to the development of various diseases, such as cancer and diabetes [67]. In the AMPK-SIRT1 pathway, AMPK enhances the expression level of SIRT1 by increasing NAD+ concentration, while SIRT1 regulates apoptosis and autophagy by increasing its transcriptional activity through catalyzing the deacetylation of FOXO1 [68]. mTOR can also be inhibited by the AMPK signaling pathway, leading to the activation of the autophagic response. It has been shown that grape seed proanthocyanidins can activate SIRT1 signaling, leading to the deacetylation of the *FOXO1* gene, thus counteracting OS in human GCs and alleviating their autophagy [69,70].

The FOXO1 pathway and GC autophagy are inextricably linked to apoptosis. FOXO1 induces autophagic death in GCs with oxidative damage, and an increase in autophagy is often seen in OS [71]. FOXO can also influence the expression of numbers of Bcl-2-family proteins such as Bim, stimulate the expression of death receptor ligands such as Fas ligands and tumor-necrosis-factor-related apoptosis-inducing ligands, and induce cell death through mitochondria-mediated endogenous pathways and death-receptor-mediated exogenous pathways [72]. Additionally, *Periplaneta americana* peptide can resist H_2_O_2_-induced apoptosis in porcine GCs via the JNK/FOXO1 pathway [73]. The relationship between autophagy and apoptosis under OS is still unclear, but it has been shown that early induction of autophagy contributes to OS-induced porcine GC apoptosis [74]. Moreover, Yang et al. [75] demonstrated that inhibition of autophagy in OS attenuated lysophosphatidylcholine (LPC)-induced apoptosis in mouse ovarian GCs. Interestingly, autophagy may also inhibit apoptosis of GC and play a protective role. Studies have demonstrated that low levels of ROS can modify Atg4 and high mobility group protein 1 (hmGB1), triggering the activation of AMPK and ASK/JNK signaling pathways, or trans-activating various proteins that can increase the expression of autophagy-related genes, ultimately resulting in decreased apoptosis [76]. Highly expressed non-coding RNA (NORHA) is another lncRNA present in GCs, exhibiting elevated expression specifically in atretic follicles. This NORHA can induce apoptosis in porcine GCs by affecting the activity of the miR-183-96-182 cluster and FOXO1 axis [77].

### 2.4. Nrf2

To counteract the adverse effects of OS generated by the external environment, the body plays a vital role in the induction of the antioxidant response through Nrf2. This key transcription factor regulates the resistance to OS and is a positive regulator of the human antioxidant response element (ARE), which drives the expression of antioxidant enzymes such as NQO1, heme oxygenase-1 (HO-1), SOD, etc. [78]. Hybertson et al. [79] indicated that the activation of Nrf2 could greatly enhance the amount of superoxide dismutase activity. The study by Esfandyari et al. [80] demonstrated that sulforaphane increased antioxidant enzymes such as SOD and CAT through an Nrf2-mediated pathway, which subsequently reduced intracellular ROS production and human GCs apoptosis. Under normal physiological conditions, Nrf2 is present in the cytoplasm associated with the negative regulatory protein Kelch-like ECH-associated protein 1 (KEAP1), which interacts with Nrf2 and acts as an adapter protein, maintaining Nrf2 at low levels and allowing its sequential degradation through the proteasome during ubiquitin-mediated processes. Dissociation from KEAP1 prevents ubiquitination of Nrf2. Upon translocation of Nrf2 to the nucleus, it forms complexes with co-activators and binds to promoter regions (AREs). This binding induces the transcription of cytoprotective genes. In addition, Nrf2 activation enhances the activity of the innate immune system, thereby attenuating or eliminating many bacterial and viral pathogens [81]. The KEAP1-Nrf2 system has emerged as an important therapeutic target for cancer, neurodegenerative diseases, and many autoimmune and inflammatory diseases [82,83,84].

Glutamine metabolism is involved in multiple redox reactions in tumor cells and affects the redox balance of cells through metabolic remodeling. Glutamine metabolism can also regulate the production and scavenging of ROS, which further affects the redox state of tumor cells, thereby affecting the stress response and survival of tumor cells. Activation of both Nrf2 and the KEAP1/Nrf2 pathway increases glutamine metabolism and promotes cell survival in KRAS-mutated pancreatic and non-small cell lung cancers, leading to greater chemoresistance [85,86]. Inhibition of glutamine metabolism can increase the chemosensitivity of cancer cells and improve the effectiveness of therapeutic drugs [41,87]. The formation of stress granules (SGs) is closely related to glutamine metabolism. SGs can regulate the redox balance, and different types of oxidative stress have different regulatory effects on the formation of SGs, which may be related to the occurrence of neurodegenerative diseases [88]. The formation of SGs may be part of the early stress response of aging cells, and the number of SGs increases significantly after aging cells are fully formed [89]. In pancreatic cancer, KRAS mutations lead to metabolic remodeling and upregulation of Nrf2, which impairs the effect of chemotherapy drugs on cancer cells. Meanwhile, controlling glutamine metabolism can inhibit the formation of SGs and increase the sensitivity of pancreatic cancer cells to chemotherapy [86].

Inhibition of nicotinamide adenine dinucleotide phosphate oxidase 4 (NOX4) expression reduces OS and apoptosis in rat GCs by activating the NRF-2/HO-1 signaling pathway, thereby alleviating PCOS in a rat model [90]. 2,2,4,4-tetrabromodiphenyl ether (BDE-47) induces OS through the Nrf2/HO-1 signaling pathway and further inhibits the expression of ovarian hormone biosynthesis-related proteins [91]. Copper ions can also cause OS in human ovarian GCs (COV434) through the Nrf2/HO-1 signaling pathway [92]. Additionally, ethanol will reduce the ability of bovine GCs to counteract OS by downregulating the expression of genes in the Nrf2 pathway signaling pathway [93].

In pharmacotherapy, vitamin E can regulate apoptosis in bovine GCs through the Nrf2-mediated PI3K/AKT and ERK1/2 pathways [40]. In contrast, genistein works through the estrogen receptor (ER)-Nrf2-FOXO1-ROS pathway to improve endocrine function and prevent oxidative damage in mouse ovaries. As a result, vitamin E exhibited therapeutic potential for PCOS [94]. Additionally, salidroside can alleviate dihydrotestosterone-induced OS in KGN cell lines by activating the AMPK/Nrf2/HO-1/NQO1 pathway [95]. AREs can also be involved in this pathway. It has also been shown that sulforaphane can protect human GCs from OS by activating the Nrf2/ARE signaling pathway [80,96]. Activation of this signaling pathway can alleviate PCOS [97]. In contrast, when bovine GCs are under OS, Nrf2-mediated signaling is the primary regulator in reducing ROS, repairing DNA, and blocking endoplasmic reticulum stress [98]. N-acetylcysteine (NAC) can improve mice GC OS by regulating the Nrf2 signaling pathway, thereby improving oocyte mitochondrial function and, thus, oocyte quality [99]. Morroniside can also regulate the Nrf2 signaling pathway and promote the nuclear translocation of Nrf2, which protects human GCs from H_2_O_2_-induced OS and inhibits apoptosis [64]. Additionally, humanin regulates OS in GCs by activating the KEAP1/Nrf2 signaling pathway, thereby alleviating PCOS [100]. Growth hormone can also ameliorate OS in the follicular fluid of advanced-stage women undergoing in vitro fertilization by affecting the expression of Nrf2/KEAP1 [101].

### 2.5. Mitophagy

Mitochondria are the primary site of cellular ROS production [102]. Under normal conditions, intracellular ROS can act as intracellular signaling molecules to regulate the normal physiological functions of the organism. When tissues are damaged, there is a large increase in ROS in the mitochondria, which damages the mitochondria and leads to abnormal mitochondrial function. Antioxidants can ameliorate damage by reducing oxidative damage to mitochondria. Mitochondrial autophagy is a selective autophagy that often occurs in defective mitochondria after mitochondrial injury or stress, and is often considered as the main mechanism of mitochondrial quality control by targeting phagocytosis and destruction of mitochondria, promoting mitochondrial turnover, and preventing the accumulation of dysfunctional mitochondria, which leads to cell degeneration [103]. It is mediated by the HIF-1α-BNIP3 signaling pathway and regulated by PINK1 and parkin proteins [104]. In OS, excessive ROS impairs HIF-1α-pathway-mediated mitophagy, and abnormalities in the mitochondrial quality control system lead to the accumulation of excess or damaged mitochondria. Under hypoxic conditions, mitophagy prevents ROS levels from increasing by removing damaged mitochondria, which attenuates death [105]. This, in turn, reduces cell death, which in part suggests that mitophagy may have a defensive role in OS [106]. Experiments that explored activation of the antioxidant defense system by *NFE2L2* and inhibition of hypoxia-induced apoptosis in porcine GCs by mitophagy also support this. *Periplaneta americana* peptide can also inhibit OS and reduce apoptosis in KGN cells by upregulating the expression levels of Bcl-2-like protein 13 (Bcl2L13) and p62 after H_2_O_2_ treatment and increasing the interaction between Bcl2L13 and LC3B, therefore increasing mitophagy and subsequently inhibiting mitochondrial damage [107]. However, it has also been shown that inhibition of mitophagy protects mouse GCs from oxidative damage [108].

### 2.6. Inflammation

The inflammatory state is closely related to the OS state, and NF-κB is the key that links inflammation to OS. AP-1 is involved in the regulation of a wide range of physiological processes including cell proliferation, apoptosis, survival, and differentiation [109,110].

ROS and intracellular redox status, especially sulfhydryl homeostasis, can directly affect AP-1 activity, and AP-1 gene expression can also be regulated by cytokines. NF-κB is a particular transcription factor that can be activated by various pathological factors and is involved in regulating the expression of many inflammatory factors. It is essential for the transcription of many pro-inflammatory genes. At the same time, NF-κB has a vital role in regulating the expression of various enzymes (e.g., iNOS, COX-2, etc.) involved in the amplification and persistence of the inflammatory response (i.e., cascade waterfall effect). Tumor necrosis factor alpha (TNF-α) can induce the production of oxygen radicals, promote the “oxidative burst” of neutrophils, produce oxygen radicals and other ROS mediators, and cause tissue damage, while ROS can also increase the level of TNF-α. TNF-α can also activate the transcription and expression of NF-κB-pathway-mediated inflammatory mediators and induce the expression of various inflammatory mediators that were discovered in vascular endothelial cells and vesicles located within the endothelial cell through the NF-κB pathway, forming an autocrine circuit that can aggravate inflammatory damage and promote the development of inflammation [111].

Transcriptome analysis of the bovine transcriptome under acute heat stress showed upregulation of inflammation-related genes, including cytochrome c (CYCS), Toll-like receptor 2 (TLR2), Toll-like receptor 4 (TLR4), and interleukin 6 (IL-6) expression, and consequently, apoptosis-related genes, suggesting that an inflammatory response occurs during OS injury and can eventually lead to apoptosis [61]. Under LPS treatment, TLR2 protein and other pro-inflammatory mediator expressions were increased in GCs and induced OS in GCs. In contrast, resveratrol can act as an anti-inflammatory agent, inhibiting OS and thus restoring GC function [112]. During the periparturient period, non-esterified fatty acids are increased in follicular fluid, leading to OS and inflammatory responses in the serum of cows via the NLRP3 inflammasome and the TLR4/NF-κB pathway. NAC can restore GC function by regulating the levels of non-esterified fatty acids, thereby reducing oxidative and free radical levels [113]. Deoxynivalenol (DON), on the other hand, can promote the inflammatory response by increasing intracellular ROS levels and upregulating TNF-α, IL-6, and IL-β [114]. Conversely, melatonin can improve OS and attenuate the inflammatory response [115].

### 2.7. Other

In addition to the classical mechanisms described above, alterations in other mechanisms may also affect OS in GCs, which in turn may lead to impaired ovarian function. For example, lower antioxidants levels reduced SOD2, which leads to elevated OS, resulting in an imbalance between ROS production and antioxidant capacity in mitochondria, ultimately leading to oxidative mitochondrial damage, reduced mitochondrial function, and increased susceptibility to apoptosis [116].

MicroRNAs (miRNAs) are small, siRNA-like molecules encoded by higher eukaryotic genomes. miRNAs degrade mRNAs or block their translation by base pairing with target gene mRNAs to guide the RNA-induced silencing complex (RISC). They play a huge role in cell differentiation, biological development, and disease progression, and they are increasingly attracting the attention of researchers. Further research on the mechanism of miRNA action and using the latest high-throughput technologies, such as miRNA microarrays, to investigate the relationship between miRNAs and diseases will lead to greater understanding of the network of gene expression regulation in higher eukaryotes. This will also enable miRNAs to become new biological markers for disease diagnosis. It may also allow this molecule to become a target for new drugs or to be copied for drug development, providing a new means of treatment for human diseases. miR-196b-5p significantly increases the expression of *CYP19A1* and *GLUT4* and decreases the levels of radixin and leucine-rich repeat-containing 17 in ovarian GCs, thereby regulating their OS, glucose uptake, and steroidogenic pathways, and then promoting follicle development and maturation [117]. miR-183-96-182, on the other hand, can be affected by NORHA and thus promote apoptosis in porcine GCs [77]. In contrast, MiRNA-146b-5p overexpression alleviated POF in mice by inhibiting the Dab2ip/Ask1/p38-Mapk pathway and γH2A.X phosphorylation [63]. Additionally, miR-30a-5p inhibits GC death by targeting Beclin1 [118].

Hypoxia-inducible factor 1α (HIF-1α) is highly expressed in GCs under H_2_O_2_-induced OS and significantly increases the level of inflammation. In contrast, chitosan oligosaccharides (COS) protect GCs from OS and apoptosis by inactivating the HIF-1α/VEGF signaling pathway [119]. Additionally, iron death can be induced by OS in GCs. In the homocysteine-induced KGN model, elevated iron death-related protein expression levels were detected. In contrast, the ferrokinase inhibitor ferrostatin-1 inhibited OS damage in GCs by regulating the levels of TET1/2 and DNA methylation [120]. Epigenetic changes can also regulate OS in GCs. 3-nitropropionic acid treatment of GCs resulted in significant downregulation of IGF2BP1. It enhanced the stable phenotype of MDM2 mRNA in an m6A-dependent manner, thereby reducing GC apoptosis and altering GC viability and cell cycle [121].

Perfluorooctanoic acid can lead to DNA damage during OS in GCs and affect the maturation of porcine GCs in vitro [122]. In contrast, nuclear shuttling of TRDMT1 accompanies the process of H_2_O_2_-induced OS in GCs. It is involved in repairing ROS-induced GC DNA damage, which provides a new idea for treating POF [123].

## 3. Potential Medication for Granulosa Cells under Oxidative Stress

During past studies, scientists discovered that there are potential drug treatment targets for these mechanisms. Figure 2 shows the mechanism by which the relevant potential medications improve GC function. Sulforaphane, *Periplaneta americana* peptide, resveratrol, astaxanthin, melatonin, celastrol, and growth hormone are particularly important. The mechanism of these drugs is also better studied. Figure 3 shows the chemical structure of some of the major medications.

### 3.1. Sulforaphane

Sulforaphane, an isothiocyanate derived from glucoraphanin, is mainly found in cruciferous plants such as radishes, cabbage, broccoli, and other vegetables [124]. Sulforaphane is a common antioxidant that has been found to affect *NFE2L2* and to regulate phase II enzymes, and its potential therapeutic effects on cancer have led to a great deal of research interest [125]. It has been suggested that sulforaphane may protect GCs of patients with PCOS from OS by activating the AMPK/AKT/Nrf2 signaling pathway and reducing the levels of ROS and apoptosis [97]. Sohel et al. [126] used bovine GCs treated with varying concentrations of sulforaphane and found that the expression of Nrf2 was markedly elevated in sulforaphane-treated GCs. The antioxidant and anti-apoptotic effects, however, were dependent on the concentration of sulforaphane used, and it was observed that high concentrations of sulforaphane (20 μM) may increase apoptosis under oxidative stress; in a follow-up study, Sohel et al. [127] also found that sulforaphane pretreatment could activate the Nrf2-ARE pathway and thus effectively protect bovine GCs from oxidative damage. Recently, Zhang et al. [106] showed that sulforaphane could activate the transcription factor of *NFE2L2* within the nucleus, which increased the expression of antioxidant enzymes and induced mitophagy, demonstrating that sulforaphane could attenuate hypoxia-induced apoptosis in GCs through the study of a sulforaphane-treated porcine GCs hypoxia model; in other experiments, sulforaphane was also seen to protect human GCs from OS by increasing the expression of antioxidant enzymes such as SOD and CAT through Nrf2 and reducing ROS production and apoptosis [80].

### 3.2. Periplaneta Americana Peptide

*Periplaneta americana* is found in tropical, subtropical, and temperate regions, and although it is generally considered a pest, it also has potential medical value [128]. Some of the pharmacological effects of *Periplaneta americana* were discovered in China thousands of years ago. It is documented in Shennong’s Herbal Classic that *Periplaneta americana* was used in traditional Chinese medicine to treat edema, burns, and various wounds and ulcers. Modern medicine has further studied *Periplaneta americana* extract and found that it possesses several outstanding biological properties, including antibacterial, antioxidant, anti-inflammatory, and anti-cancer properties [129,130]. It has been demonstrated that *Periplaneta americana* peptide can significantly inhibit H_2_O_2_-induced apoptosis in KGN cells and porcine ovarian GCs by upregulating SIRT1 expression and reducing FOXO1 expression [73,131,132]. Moreover, *Periplaneta americana* peptide can also inhibit mitochondrial damage by upregulating Bcl2L13-mediated mitochondrial autophagy, thereby suppressing OS-induced apoptosis in KGN cells [107].

### 3.3. Resveratrol

Resveratrol is a polyphenolic compound, mainly derived from peanuts, grapes (red wine), thuja, mulberry, and other plants, with antioxidant, anti-inflammatory, and anti-cancer properties, as well as protective effects on the central nervous system and cardiovascular system [133,134,135,136,137]. Moreira-Pinto et al. [138] treated human GCs with different concentrations of resveratrol at different times and found that low doses of resveratrol had the ability to alleviate OS in GCs. Resveratrol has been shown to reduce OS and inhibit GC apoptosis in a POF rats model by activating the PI3K/AKT/mTOR signaling pathway; it can also minimize LPS-induced inflammation and OS in PCOS GCs by inhibiting the expression of TLR2 [112,139]. Moreover, resveratrol is believed to be a SIRT1 inducer that increases antioxidant enzyme activity, reduces ROS levels, and decreases apoptosis [140].

### 3.4. Astaxanthin

Astaxanthin is a type of carotenoid, a ketone compound, a natural super antioxidant molecule found in shrimp, crab, salmon, green algae *Haematococcus pluvialis*, red yeast, and other marine organisms [141]. Astaxanthin has the potential to prevent and treat liver disease and cardiovascular disease, and demonstrates anti-diabetes and anti-cancer as well as skin-protecting properties [142,143,144,145,146]. Studies have shown that astaxanthin enhances the development of oocytes grown in vitro [147], and improves antioxidant capacity of follicles and oocytes and reduces bisphenol A (BPA)-induced OS [148]. In a randomized controlled trial conducted by Gharaei et al. [149], it was discovered that treating patients with polycystic ovaries with astaxanthin led to activation of the Nrf2 axis and increased antioxidant capacity in GCs, indicating its potential in attenuating oxidative stress response in such individuals. A study by Eslami et al. [150] also found that the protective effect of astaxanthin on oxidatively stressed cells was mediated by Nrf2 activation of upregulated phase II enzymes.

### 3.5. Melatonin and Celastrol

Melatonin is a methoxyindole produced by the pineal gland, and the photoperiod governs its synthesis and secretion with a circadian rhythm [151]. In addition to improving sleep, melatonin has antioxidant, anti-aging, immunomodulatory, and anti-cancer biological functions [152]. Moreover, melatonin can protect the brain and treat cardiovascular diseases [153,154]. Additionally, it has been demonstrated that as an antioxidant, melatonin can reduce oxidative damage in follicles and improve oocyte quality in infertile women [155]. It has also been shown to attenuate ovarian GC loss in young women treated with cyclophosphamide chemotherapy by inhibiting the mitochondrial apoptosis pathway [45]. Fan et al. [114] found that deoxynivalenol (DON) exposure increased OS in mouse ovarian GCs, whereas melatonin ameliorated OS, mitochondrial dysfunction, and inflammation. Experiments by Xue et al. [156] revealed that melatonin protects human GCs from the detrimental effects of DON through inhibition of ER stress and FOXO1. Melatonin’s defense against OS in mouse GCs can also be achieved by inhibiting the PINK1-Parkin pathway and thus mitochondrial autophagy [108].

Celastrol is a bioactive compound isolated from the roots of *Tripterygium wilfordii*. Modern research has shown that it has outstanding efficacy in anti-cancer studies and protective effects against a wide range of cardiovascular diseases, diabetes, and other metabolic diseases [157,158,159]. Interestingly, celastrol, like melatonin, improves the survival of human granulosa-lutein cells under OS. The difference is that melatonin induces *SIRT1*, *SIRT6*, and *SIRT7* gene expression, while celastrol only induces *SIRT7* gene expression [160].

### 3.6. Growth Hormone

Growth hormone is a peptide hormone secreted by the human anterior pituitary gland that affects bone formation, promotes protein synthesis, regulates fat and carbohydrate metabolism, and plays a crucial role in human growth and development [161,162]. Growth hormone plays a role in maintaining the structure and function of the average adult heart, and treatment with growth hormone may improve lipid profiles and benefit blood vessel walls [163]. In rats treated with azidothymidine, growth hormone treatment also promotes hematopoietic recovery and reduces myelotoxic effects [164]. In addition, it can affect fertility [165]. Growth hormone has been shown to promote oocyte maturation by promoting meiotic progression, balancing redox homeostasis in the microenvironment, and enhancing oocyte development [166]. Growth hormone can also improve implantation rates and pregnancy productivity in patients with poor prognoses undergoing IVF by directly affecting oocyte and embryo quality [167]. This may be related to the expression of Nrf2/KEAP1 in GCs [101]. It has been shown that pretreatment with growth hormone reduces ROS levels in GCs and attenuates OS in poor ovarian responders [168]. Feng et al. [169] have similarly concluded that recombinant human growth hormone attenuates OS and GC apoptosis induced by ROS and mitochondrial superoxide. Recently, growth hormone was shown to reduce OS and enhance mitochondrial function through the SIRT3-SOD2 pathway, thereby decreasing cisplatin-induced KGN cell apoptosis [170]. It has also been shown to reduce OS-induced apoptosis in GCs of patients with PCOS by activating the PI3K/AKT signaling pathway [50].

In addition to the above drugs, some other drugs may also be able to inhibit OS in GCs by targeting the above targets. Table 1 summarizes the mechanisms and effects of some drugs on OS in GCs.

## 4. Conclusions and Perspectives

Thus, OS in GCs can lead to various female reproductive disorders, with POF and PCOS in particular being the most common, in addition to diseases associated with reduced ovarian functional reserve. OS within GCs leads to significant cellular phenotypic alterations, such as apoptosis and autophagy. In addition, this process is accompanied by other alterations, such as decreased antioxidant levels, impaired steroid synthesis, and mitochondrial dysfunction. Related reports suggest that the causes of OS in GCs are classified into various causes, including changes in environmental factors, chemical toxicants such as BPDE and nonylphenol, and heavy metal contamination, which can lead to OS in GCs. The specific mechanism of OS in GCs still needs further study. Existing studies have shown that other signaling pathways such as the PI3K/AKT signaling pathway, MAPK signaling pathway, FOXO axis, Nrf2/KEAP1 signaling pathway, inflammation-related signaling pathway, and mitophagy are involved in the process of OS in GCs.

Current studies suggest that drugs such as sulforaphane, *Periplaneta americana* peptide, resveratrol, astaxanthin, melatonin and ryanodine, and growth hormone can alleviate some female reproductive disorders by improving these signaling pathways and thus inhibiting OS in GCs. However, the efficacy studies of most of these drugs are still in the laboratory stage, and a lot of animal experiments and clinical data are still needed to support their clinical application. Future randomized controlled clinical trials on humans are necessary to demonstrate the efficacy and safety of drug therapies which focus on PCOS and POF. We hope this paper can provide new ideas and suggestions for future experimental studies and clinical applications.

## Figures and Tables

**Figure 1 ijms-24-09205-f001:**
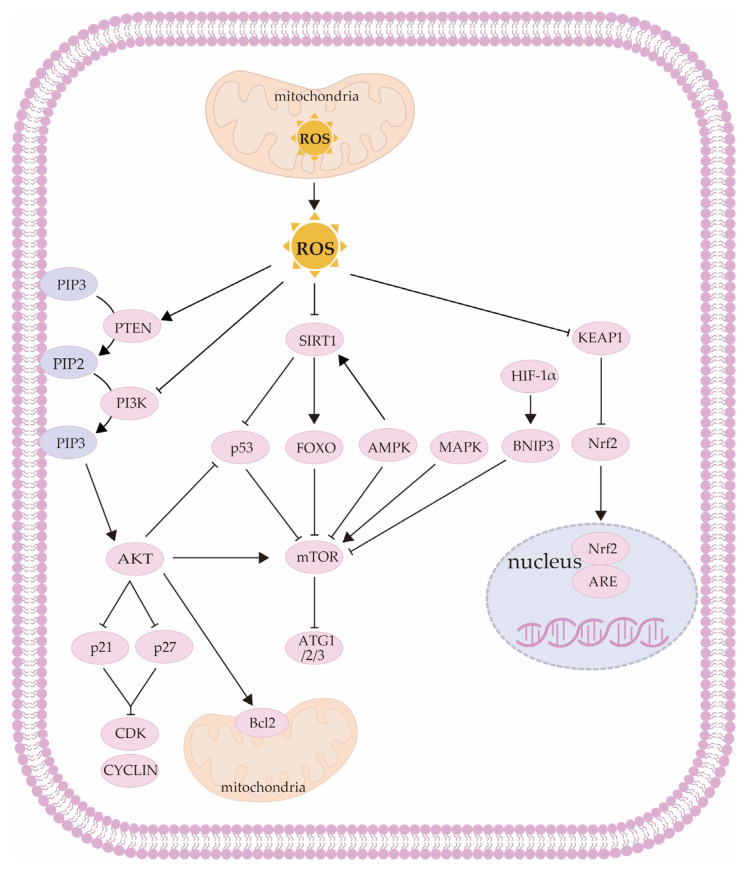
Schematic diagram of role of ROS in GCs. AKT: protein kinase B; AMPK: AMP-activated protein kinase; ARE: antioxidant response element; ATG1/2/3: autophagy-related protein 1/2/3; Bcl-2: B-cell lymphoma 2; BNIP3: Bcl-2/adenovirus E1B 19-kDa-interacting protein 3; CDK: Cyclin-dependent kinase; FOXO: forkhead box O; HIF-1α: hypoxia-inducible factor 1 subunit α; KEAP1: Kelch-like ECH-associated protein 1; MAPK: mitogen-activated protein kinase; mTOR: mammalian target of rapamycin; Nrf2: nuclear factor erythroid 2-related factor 2; p21: Cyclin-dependent kinase inhibitor p21; p27: Cyclin-dependent kinase inhibitor p27; p53: Cyclin-dependent kinase inhibitor p53; PI3K: phosphatidylinositol-3 kinase; PIP2: phosphatidylinositol bisphosphate; PIP3: phosphatidylinositol (3,4,5) trisphosphate; PTEN: phosphatase and tensin homolog; ROS: reactive oxygen species; SIRT1: silent information regulator sirtuin 1.

**Figure 2 ijms-24-09205-f002:**
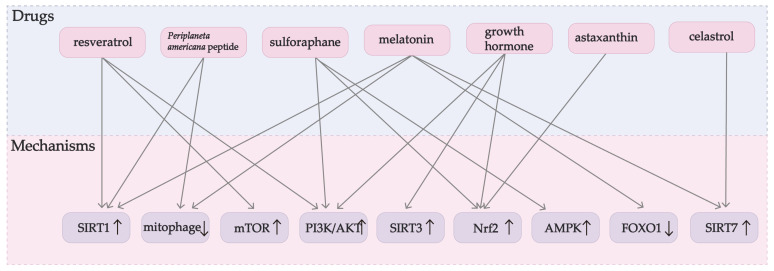
Schematic diagram of each medication’s mechanism to relieve OS in GC. AKT: protein kinase B; AMPK: AMP-activated protein kinase; FOXO: forkhead box O; MAPK: mitogen-activated protein kinase; mTOR: mammalian target of rapamycin; PI3K: phosphatidylinositol-3 kinase; SIRT1: silent information regulator sirtuin 1; SIRT3: silent information regulator sirtuin 3; SIRT7: silent information regulator sirtuin 7.

**Figure 3 ijms-24-09205-f003:**
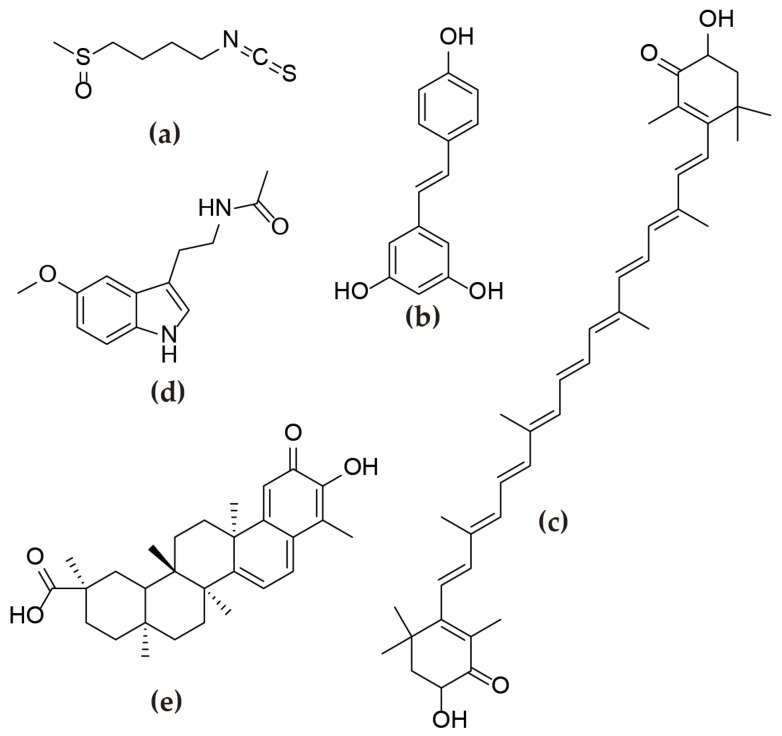
Chemical structures of main medications: (**a**) sulforaphane; (**b**) resveratrol; (**c**) astaxanthin; (**d**) melatonin; and (**e**) celastrol.

**Table 1 ijms-24-09205-t001:** Mechanisms and effects of each medication on its model.

Ingredient	Model	Mechanism	Effect
Sulforaphane [80,96,97,106]	Porcine GCs	Increases antioxidant capacity and mitophagy through *NFE2L2*.	Inhibits granule cell apoptosis.
	Human GCs	Upregulates the expression of antioxidants through the Nrf2/ARE pathway.	Reduces ROS production and apoptosis.
	Human granulosa-lutein cells	AMPK/AKT/Nrf2 pathway.	Reduces ROS and apoptosis levels.
*Periplaneta americana* peptide [73,107,132]	KGN cells	Increases the expression of Bcl2L13 and its interaction with LC3B.	Promotes mitophagy and reduces apoptosis.
	Pig ovary GCs	FOXO1 pathway.	Reduces apoptosis.
Resveratrol [112,138,140,171]	PCOS cellular model	Inhibits the expression of TLR2 in GCs.	Alleviates inflammation and OS.
	Female mice		Alleviates BPA-induced GC OS, autophagy, and apoptosis levels.
	Human luteinized GCs	Increases antioxidant enzyme activity through the Sirt1.	Reduces apoptosis rate and ROS level.
	Human GCs	Reduces ROS/RNS formation after OS.	Protects GCs.
Astaxanthin [148,149,150]	PCOS patients	Activates the Nrf2 axis.	Upregulates antioxidant levels and total antioxidant capacity.
	Mouse follicles		Improves the antioxidant capacity and reduces OS in follicles and oocytes.
	Human GCs	Activates Nrf2/ARE pathway.	Inhibits ROS production and cell death.
Melatonin [45,108,114,156,160]	Mouse GCs	Inhibits mitophagy through the MEL-PINK1-Parkin pathway.	Reduces GC death.
	Female mice		Reduces ovarian GC apoptosis and maintains AMH expression.
	Murine ovary GCs	Antioxidant and anti-inflammatory effects.	Reduces oxidative damage.
	Human GCs	Reduces ER stress and inhibits the FOXO1 pathway.	Alleviates GC apoptosis.
	Mice and mouse ovarian granule cells		Inhibits excessive OS and apoptosis.
	Human granulosa-lutein cells	*SIRT1*, *SIRT6*, and *SIRT7* gene expression.	Improves cell viability.
Celastrol [160,172]	Human granulosa-lutein cells	Induces *SIRT7* gene expression.	Improves cell viability.
	Human granulosa-lutein cells		Regulates human granulosa-lutein cells gene expression and regulates OS imbalance.
Growth hormone [50,101,168,169,170,173]	Rats with POI	Promotes the balance between OS and cellular oxidant detoxification.	Alleviates OS and apoptosis of GCs.
	KGN cells	SIRT3-SOD2 pathway.	Reduces OS and enhances mitochondrial function, and inhibits apoptosis of GCs.
	Older women undergoing IVF	Nrf2/KEAP1 pathway.	Alleviates OS in follicle fluid.
	Poor ovarian responders		Reduces OS by improving antioxidant capacity and reducing ROS.
	PCOS patients		Improves mitochondrial dysfunction and relieves OS.
	PCOS patients	Activates the PI3K/AKT pathway.	Reduces ROS levels and apoptosis.
Vitamin D [117]	Mouse ovaries	miR-196b-5p.	Regulates OS and hormone synthesis of GC.
Vitamin E [40]	Bovine GCs	Upregulates Nrf2-mediated defense system by activating the PI3K/AKT and ERK1/2 pathways.	Promotes GC proliferation and inhibits apoptosis.
Tilapia skin peptides [174]	Mouse with POF	Modulates the Bcl-2/BAX/CASP-3 apoptosis pathway and enhances the Nrf2/HO-1 pathway.	Attenuates OS and apoptosis.
Grape seed proanthocyanidins [69]	Chicken follicular GCs	Inhibits FOXO1 and activates the PI3K-AKT pathway.	Reduces GCs’ autophagy and oxidative damage.
Morroniside [44,64]	Rat ovarian GCs	PI3K/AKT/mTOR pathway.	Reduces apoptosis and autophagy of rat GCs.
	Human GCs	Regulates the Nrf2 and MAPK pathways.	Inhibits GC apoptosis.
Humanin [100,175]	KGN cells		Increases cell viability and reduces apoptosis.
	PCOS patients	Regulates the KEAP1/Nrf2 signaling pathway.	Alleviates OS.
Metformin [52]	Rats with PCOS	PI3K/AKT/mTOR pathway.	Reduce autophagy and OS.
Kurarinone [176]	KGN cells	Activates the PI3K/AKT pathway.	Alleviate OS and apoptosis.
Isorhamnetin [55]	Porcine GCs	Activates the PI3K/AKT pathway.	Promotes cell proliferation, inhibits apoptosis, and regulates hormone synthesis.
Chrysin [177]	Mice ovarian GCs		Reduces inflammation and OS.
Baicalin [54]	Human GCs/mice ovaries/older mice	mTOR pathway.	Enhances the viability and viability of granule cells.
Dietary flavonoid isoquercitrin [178]	Human ovarian GCs HGL5	Inhibits ROS production.	Reduces OS.
Ginsenoside Rb1 [179]	Ovarian GCs from women and mice	Activates AKT phosphorylation and enhances AKT-FOXO1 interaction.	Inhibits OS.
Genistein [94]	Mice with PCOS	Increases the expression of Nrf2 and FOXO1 through the ER-Nrf2-FOXO1-ROS pathway.	Alleviates OS.

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
