# Peer review of "Mechanisms of and Potential Medications for Oxidative Stress in Ovarian Granulosa Cells: A Review"

_ijms, 2023, doi:10.3390/ijms24119205_

Round 1
Reviewer 1 Report
A timely review article by Dr. Pan and the group discussed oxidative stress's role in ovarian granulosa and mentioned the possible therapeutic opportunities. This is a very well-written review article that impacts the translational aspect of the basic research. A few things must be addressed before it is ready for acceptance. They are as follows:
1. Mutant RAS is known to be responsible for ovarian cancer (15%) PMID: 33870211 and PMID: 36451660.
Authors should discuss this aspect by adding a few lines in the context of the current manuscript.
2. It has been shown that NRF2 plays a significant role in regulating glutamine metabolism PMID: 31911550 and PMID: 31040157. Glutamine metabolism also mediates the redox status of the cancer cells PMID: 33870211 and PMID: 25476909. The authors must add a few lines discussing this point.
3. Authors also should add another novel aspect of stress response - stress granule formation PMID: 28194255 and PMID: 19176530. It has been postulated that stress granule formation is also associated with glutamine metabolism PMID: 31911550, which links up with redox biology. Authors should mention this topic as one of the aspects where further studies are warranted.
Reviewer 2 Report
The work is a review of the mechanisms and signaling pathways that oxidative stress induces in granulosa cells and how various drugs/antioxidants affect these alterations. It is an original work that has not been proposed before. The review is exhaustive and contains updated bibliography on the subject. However, several aspects of the manuscript need to be corrected.
- There are bibliographic references that have nothing to do with what is being presented (see, for example, citations 41, 42, 56, 92, 114, or 121).
- There is literature that refers to effects that have been described in cells other than granulosa cells, such as cumulus cells (with different functions and metabolism), or refers to cell lines, also with different metabolism (see, for example, citations 2, 10, or 106).
- Other references related to signaling mechanisms and pathways refer to processes such as aging or cancer, and not to infertility or reproductive diseases (see refs. 11, 40, 41, 41, 67, 68).
- On numerous occasions certain indicators and abbreviations are used, many of which have been obtained directly from the literature, without reference to what they mean or the role they play (and are outside the context of the article cited). Abbreviations that have not been previously defined are also used.
See, for example, "m6A", line 148; "CASP3", line 158; "PM2.5", line 136; "BPDE", line 144; "B(a)D" line 145; "BALB C PCOS", line 163; "BPA", line 459; "KGN", line 269; "RDX and LRRC17", line 367; "KK-1", line 145; GC, GCs, ROS (which is first defined in line 504), OS... On the other hand, abbreviations that are only named once should be omitted.
- Abbreviations should be described the first time they are cited, and the abstract should not contain abbreviations.
- It is desirable that descriptions of findings found in the literature indicate the biological model used, as well as their origin (pig, chicken, mouse, rat,...).
- Figures 1 and 2. Write correctly the protein abbreviations and include a legend, defining abbreviations.
- There is ample evidence that certain reproductive diseases are associated with oxidative stress. However, the presence of oxidative stress markers in the blood of women with PCOS does not necessarily reflect OS in granulosa cells. Paragraph 352-355 is superfluous.
- English needs revision.
- Review sentences like:
- "And BPDE is a metabolite from B(a)P that affects KK-1 in granulosa cells, affecting the luteinization process". Line 145. (KK-1 is a granulosa cell line).
- "To counteract antioxidant effects, astaxanthin and metformin treat BALB C PCOS by increasing Akt expression in granulosa cells [54]. And baicalin improved granulosa cell function in aged mice through the PI3K/Akt/mTOR pathway, upregulated Bcl-2 and downregulated Bax and caspase 3, thereby inhibiting apoptosis and increasing the secretion of steroid hormones, thereby restoring ovarian function [55,56]". Lines 163-166.
- "it has been shown that low levels of ROS modifying Atg4 and high mobility group protein 1 (hmGB1) protein, activating AMPK and ASK/JNK pathways, or trans-activating various proteins that can upregulate autophagy, leading to reduced apoptosis [78]. There is another lncRNA in granulosa cells called highly expressed non-coding RNA (NORHA), which is highly expressed in atretic follicles". Lines 229-232
- "Sohel MMH et al. [125] treated bovine GCs with different concentrations of SFN in their experiments, and the experimental results showed that the expression of Nrf2 was significantly increased in SFN-treated GCs, while the antioxidant and apoptotic effects on GCs were concentration-dependent, and high concentrations of SFN (20 μM) may exacerbate apoptosis under oxidative stress", lines 411-416.
- "Gharaei et al. [148] found that AST treatment has been shown to activate the Nrf2 axis and enhance the antioxidant capacity of granulosa cells by conducting a randomized controlled trial of astaxanthin affecting the oxidative stress response of granulosa cells in patients with polycystic ovaries;" lines 459-462.
- "Thereby exhibiting therapeutic potential for polycystic ovary syndrome [91]". Line 267
- "For example, lower antioxidants levels." Line 347
- There are numerous errors:
- Hidroxyl radical (line 58), phosphatydylinositol-3 phosphokinase line 107; FoxOs line113, ROSs line 57;
- Several names are not case sensitive, and there is confusion between genes and proteins. Examples: BCL2, P21, P53, cd (line138), FOXO1, FoxO1. "peptide, Resveratrol, Astaxanthin, Melatonin, Celastrol, and Growth hormone", line 396,
- Species nomenclatures should be in italics (Tripterygium wilfordii, Periplaneta americana, ..)
- "PCOS rats", line84
- "Nrf2/ARE Pathway", line 92
- "caspase3", line 151
- "expression of MicroRNA-146b-5p", line 186
- "attenuated Lysophosphatidylcholine" line 227
- "NF-KB"
English needs revision.
Round 2
Reviewer 1 Report
All concerns have been addressed, ready for acceptance.
Author Response
Reviewer 1:
We would like to express our sincere gratitude for the reviewer's valuable feedback regarding our manuscript. We are pleased to report that we have thoroughly addressed all the concerns raised and made appropriate revisions. We truly appreciate the time and effort the reviewer has taken in providing constructive criticism, which has undoubtedly improved the quality of our work. Thank you again for your insightful comments, and we are ready for acceptance of our revised manuscript.
Reviewer 2 Report
Revision 2
The revision has resulted in a significant improvement in the article. Some corrections will still need to be made, as indicated:
The names of several proteins should not be in capital letters. Thus, the correct names of several proteins are: "Bcl-2", "p21", and "Nrf2". Please, review them throughout the text.
Line 82. Indicate the population group with such prevalence.
Line 111. Correct "PIP2: phosphatidylinositol bisphosphate"
Line 179. Correct the sentence. ("To counteract OS").
Line 232. Delete "GSPs".
Line 248. Delete "reactive oxygen species" (already described).
Lines 293-295. I wonder if rats routinely develop PCOS. It is better to include the source as "in a rat model of PCOS".
Line 308. Correct "NEF2/ARE signaling"
Line 337. Correct "Experiments that explored activating (activation of) the antioxidant defense system by NFE2L2 and inhibiting (inhibition of) hypoxia-induced apoptosis in porcine GCs by mitophagy also support this".
Lines 357-364. Review: the paragraph and the meaning of "thylakoid". Please, include a reference.
Lines 365-369. Please, include a reference.
Line 411. Delete "MMH"
